# Pro-Inflammatory and Anti-Inflammatory Salivary Cytokines in Breast Cancer: Relationship with Clinicopathological Characteristics of the Tumor

**Lyudmila V. Bel'skaya [1],\*** [ID]**, Alexandra I. Loginova [2] and Elena A. Sarf [1]**

[1] Biochemistry Research Laboratory, Omsk State Pedagogical University, 14, Tukhachevsky Str, Omsk 644099, Russia

[2] Clinical Oncology Dispensary, 9/1, Zavertyayeva Str, Omsk 644013, Russia

\* Correspondence: belskaya@omgpu.ru

**Abstract:** The aim of the work was to compare the salivary cytokine profile of breast cancer patients with the clinicopathological characteristics of the tumor. The study included 113 patients with breast cancer (main group, mean age 54.1 years) and 111 patients with breast fibroadenomas (control group, mean age 56.7 years). Before treatment, saliva samples were collected from all patients and the content of cytokines (IL-1β, IL-2, IL-4, IL-6, IL-8, IL-10, IL-18, MCP-1, and TNF-α) was determined. The content of cytokines in saliva correlates well with the clinicopathological characteristics of breast cancer. The level of all salivary cytokines increases at advanced stages of breast cancer and at a low degree of tumor differentiation. The exception is MCP-1, for which there is an extremely high content for well-differentiated breast cancer. A statistically significant increase in the content of MCP-1, IL-1β, IL-2, IL-4, and IL-10 was found in triple-negative breast cancer. For the first time, the correlation of salivary levels of TNF-α, IL-1β, and IL-6 with HER2 status, MCP-1, IL-1β, IL-2, and IL-4 with the hormonal status of the tumor was shown. The relationship between the level of IL-2, IL-10, and IL-18 in saliva with the level of Ki-67 expression has been established.

**Keywords:** saliva; pro-inflammatory and anti-inflammatory cytokines; chemokine; breast cancer

## 1. Introduction

Cytokines are small secreted proteins that are key modulators of inflammation. There are a large number of studies showing that the expression of various cytokines is altered in breast cancer [1–5]. In particular, in diseases of the mammary gland, cytokines can be involved in the infectious–inflammatory process and allergic response at the level of immune mechanisms and the effector link, which largely determines the direction, severity, and outcome of the pathological process [3,5]. According to the mechanism of action, cytokines can be conditionally divided into pro-inflammatory (IL-1β, IL-2, IL-6, IL-12, TNF-α, chemokines—IL-8, MCP-1, and others), which are produced and act on immunocompetent cells, initiating an inflammatory response, and anti-inflammatory (IL-4, IL-10, TGF-β, etc.), which regulate specific immune responses and limit the development of inflammation [6]. Pro-inflammatory cytokines usually regulate the growth, activation, and differentiation of immune cells, as well as the direction of immune cells to infection sites in order to control and destroy intracellular pathogens, including viruses [7]. IL-1β plays an important role in inflammation and the immune response. IL-2 triggers the immune response and activates factors involved in antiviral, antibacterial, and antitumor protection, and stimulates the proliferation and activation of natural killers and cytotoxic lymphocytes. IL-4 is responsible for the humoral immune response, has local antitumor activity, suppresses the production of inflammatory cytokines (IL-8, TNF-α), and also regulates multiple biological processes, such as proliferation, differentiation, and apoptosis in various cell types [8]. IL-6 plays a key role in the development of inflammation and the

immune response to infection or tissue damage [9]. The relationship between the level of IL-6 and the early progression of breast cancer has been shown [10,11]. In general, plasma levels of IL-6 and IL-8 positively correlate with the stage of the disease and mortality from breast cancer [12]. The action of IL-8 is multidirectional: on the one hand, IL-8 causes the accumulation of neutrophils that can directly kill cancer cells; on the other hand, due to the angiogenic activity of IL-8, it promotes the formation of new vessels, which can lead to tumor development [13]. IL-8 is an important inflammatory factor, and many studies show that inflammation in the tumor microenvironment plays an important role in the progression of breast cancer [14]. IL-10 plays an important role in the pathogenesis of cancer: the excessive production of IL-10 increases the likelihood of tumors due to immunosuppression [6,15–17]. Elevated levels of IL-18 in the serum of patients with various forms of cancer correlate with the progression of the disease and the development of metastasis. On the one hand, IL-18 enhances Fas-ligand expression on natural killer cells and T-lymphocytes and inhibits the growth of blood vessels in tumor tissue. On the other hand, it stimulates the production of chemokines such as IL-8, which can provoke tumor metastasis [6]. The level of TNF-$\alpha$ is directly dependent on the severity of endogenous intoxication [18]. It is known that in breast cancer, a high level of monocytic chemoattractant protein 1 (MCP-1, CCL2) in the tumor tissue and in the circulating blood correlates with an unfavorable prognosis [19,20].

It is known that a number of cytokines, for example, IL-6, can be more easily detected in saliva than in serum or plasma [21,22]. Since cytokines found in saliva can accumulate over time, this allows for more efficient determination of their levels than those circulating in the blood [23,24]. Nevertheless, a number of studies show that it is difficult to establish an unambiguous correlation between the level of cytokines in the blood and saliva [25–27]. In most studies, the authors only pay attention to diseases of the oral cavity and periodontium [28,29]. Previously, we showed the possibility of detecting cytokines in saliva in normal individuals and in lung cancer [30].

The advantages of saliva compared to venous or capillary blood are the non-invasiveness of collection and the absence of the risk of infection when obtaining a biomaterial [31–33]. At the same time, saliva is potentially a more informative medium for its use in clinical laboratory diagnostics, including cancer diagnostics [34–36].

Previously, we have shown a change in the content of a number of cytokines in breast cancer, but the comparison was made with healthy volunteers [37]. It was shown that, compared with healthy controls, in the saliva of breast cancer patients, there is an increase in the content of both pro-inflammatory (IL-2, IL-6, and IL-18) and anti-inflammatory cytokines (IL-4 and IL-10). In the present study, we compared the content of a broader range of cytokines and chemokines in the saliva of breast cancer patients and fibroadenomas to account for the impact of local inflammation on the results. The aim of the work was to compare the salivary cytokine profile of patients with breast cancer with the clinicopathological characteristics of the tumor.

## 2. Materials and Methods

### 2.1. Study Design and Description of Study Groups

The study included 113 patients with breast cancer (main group, mean age 54.1 years) and 111 patients with breast fibroadenomas (control group, mean age 56.7 years). The study included patients with a normal body mass index (18.5–25.0) without clinically significant comorbidities, including cancer of other locations or diabetes mellitus. The inclusion in groups occurred in parallel. The following inclusion criteria were considered: (1) the age of patients 30–75 years; (2) the absence of any treatment at the time of inclusion in the study, including surgery, chemotherapy, or radiation; and (3) the absence of signs of active infection (including purulent processes). The exclusion criterion was the absence of histological or cytological verification of the diagnosis.

In all patients of the main group, invasive breast carcinoma of the following stages was histologically and cytological confirmed: in situ—7 (6.2%), stage I—34 (30.1%), stage IIa—29

(25.7%), stage IIb—18 (15.9%), stage III—14 (12.4%), and stage IV—11 (9.7%). In 70 patients, there were no signs of regional lymph node metastases ($N_0$—61.9%), in 24 patients, metastases were detected in displaced axillary lymph nodes on the affected side ($N_1$—21.2%), in 8 patients, more than two lymph nodes were affected ($N_2 + N_3$—7.1%), and distant metastases were identified in 11 patients ($M_1$—9.7%). Breast tumors were classified according to the degree of tissue differentiation into highly differentiated (G1, $n = 8$), moderately differentiated (G2, $n = 44$), and low differentiated (G3, $n = 20$). In all cases, the status of HER2, estrogen, and progesterone receptors was determined. HER2-negative status was confirmed in 80 patients (76.2%) and HER2-positive in 25 patients (25.8%); ER-negative status was confirmed in 31 patients (29.5%) and ER-positive in 74 patients (70.5%); PR-negative status was confirmed in 32 patients (30.5%) and PR-positive in 73 patients (69.5%). In eight patients, an immunohistochemical examination was not performed for a number of reasons. Ki-67 expression values less than 20% (Ki-67 low) were determined in 30 patients (32.3%) and more than 20% (Ki-67 high) in 63 patients (67.7%). By molecular biological subtypes of breast cancer, the patients were distributed as follows: triple-negative—12 (11.4%), luminal A-like—23 (21.9%), luminal B-like (HER2-negative)—15 (14.3%), luminal B-like (HER2-positive)—51 (48.6%), and non-luminal—4 (3.8%). In patients of the comparison group, the presence of fibroadenomas (single or multiple) of the mammary glands was confirmed.

## 2.2. Determination of Expression of Receptors for Estrogen, Progesterone, HER2, and Ki67

The Allred Scoring Guideline was used to assess the expression level of estrogen receptors (ER) and progesterone (PR) [38]. The calculated integrative indicator allows us to define the case under study in one of four main groups: a group with an expression level of 0 points (complete absence of stained nuclei, indicated by "−"), a group with a weak color level (index from 2 to 4 points, indicated by "+"), a group with an average level of expression (index from 5 to 6 points, indicated by "++"), and a group with a high level of expression (index from 7 to 8 points, indicated by "+++"). When determining one of the four categories of the receptors for estrogen, progesterone, and HER2 expression levels (−, +, ++, and +++), the recommendations of ASCO/CAP were followed [39]. An HER2 status assessed as "−" or "+" was considered negative, while a status assessed as "+++" was considered positive. The determination of HER2 expression was carried out using the immunohistochemical method, with an indeterminate result (++), and to confirm the HER2 status, a study was carried out using in situ hybridization (FISH) [39]. Additionally, breast cancer sub-classification differentiates these tumors into five groups: triple-negative (TN), luminal A-like, luminal B-like (HER2-negative), luminal B-like (HER2-positive), and non-luminal. The determination of the molecular biological subtype was carried out as standard by a combination of the status of HER2, estrogen, and progesterone receptors and the level of Ki-67 [40].

## 2.3. Determination of Content of Salivary Cytokines

All participants had saliva sampling in the amount of 1 mL before the start of treatment. The saliva samples were collected in the morning on an empty stomach by spitting into sterile polypropylene tubes, centrifuged at $10,000 \times g$ for 10 min (CLb-16, Moscow, Russia). The content of the cytokines in the saliva (IL-1β, IL-2, IL-4, IL-6, IL-8, IL-10, IL-18, MCP-1, and TNF-α) was determined by enzyme-linked immunosorbent assay using Vector Best kits (Novosibirsk, Russia). In all cases, the same volumes of saliva aliquots (100 μL) were used. We used calibration and control samples based on human blood serum, certified against cytokine and chemokine standards (R&D Systems, Inc., Minneapolis, USA), according to the instructions given in the corresponding reagent kits without changes, including reagent volumes and incubation time. Reading was performed using Thermo Scientific Multiskan FC (Waltham, MA, USA).

### 2.4. Statistical Analysis

Statistical analysis of the obtained data was performed using the Statistica 13.0 (Stat-Soft, Tulsa, OK, USA) software by a non-parametric method using the Wilcoxon test in dependent groups, and the Mann–Whitney U-test in independent groups. The sample was described by calculating the median (Me) and the interquartile range in the form of the 25th and 75th percentiles [LQ; UQ]. Differences were considered statistically significant at $p < 0.05$.

### 3. Results

It was shown that in the group of breast cancer patients, only the content of IL-2 and IL-10 changed statistically significantly (Table 1), and in both cases, there was an increase in concentration. For other cytokines and chemokines, an ambiguous pattern of changes in concentrations was observed. Thus, among the pro-inflammatory cytokines in breast cancer, the content of IL-1β, IL-2, and IL-6 increased (+16.1%, +25.0%, and +12.7%, respectively), while for TNF-α, MCP-1, IL-8, and IL-18, the content decreased (−36.2%, −23.8%, −15.0%, and −12.1%, respectively). Among anti-inflammatory cytokines, the same trend was noted: the content of IL-4 decreased (−15.1%), and IL-10 increased in breast cancer (+24.5%, Table 1).

**Table 1.** Content of cytokines in saliva in breast cancer.

| Indicators | Control, *n* = 111 | Breast Cancer, *n* = 113 | *p*-Value |
|---|---|---|---|
| Age, years | 54.1 [47.9; 56.7] | 56.7 [49.0; 64.0] | 0.4357 |
| Proinflammatory | | | |
| TNF-α, pg/mL | 1.74 [0.86; 2.75] | 1.11 [0.69; 2.55] | 0.2619 |
| MCP-1, pg/mL | 25.05 [7.01; 87.65] | 19.10 [6.97; 103.45] | 0.9766 |
| IL-1β, pg/mL | 185.8 [56.6; 377.7] | 215.8 [100.6; 385.8] | 0.6906 |
| IL-2, pg/mL | 1.40 [0.86; 1.88] | 1.75 [1.03; 2.73] | 0.0100 * |
| IL-6, pg/mL | 1.65 [0.94; 5.07] | 1.86 [0.92; 3.74] | 0.9295 |
| IL-8, pg/mL | 115.0 [36.8; 201.5] | 97.7 [50.1; 144.5] | 0.7211 |
| IL-18, pg/mL | 44.5 [20.3; 99.5] | 39.1 [14.9; 101.0] | 0.8110 |
| Anti-inflammatory | | | |
| IL-4, pg/mL | 0.86 [0.37; 1.57] | 0.73 [0.40; 1.60] | 0.7231 |
| IL-10, pg/mL | 1.88 [1.04; 2.42] | 2.34 [1.35; 3.19] | 0.0185 * |

Note. *—differences between groups are statistically significant, $p < 0.05$.

Due to the fact that the group of breast cancer patients was heterogeneous, at the next stage, we considered the change in the content of salivary cytokines depending on the stage of the disease. It was found that for IL-1β and IL-2, there was an increase in the content, pronounced in the early stages of the disease, and then tending to decrease and statistically significantly increasing in more advanced stages (Figure 1). For IL-6, a general upward trend was also noted; however, the nature of the change in content from stage to stage was ambiguous. The content of IL-8 and IL-18 decreased and deviations from the general trend were observed in the early stages of the disease (Figure 1). For TNF-α and MCP-1, a decrease in concentration was observed in the early stages and a subsequent increase for advanced stages. The change in the concentrations of anti-inflammatory cytokines IL-4 and IL-10 was multidirectional, with the maximum differences being observed for stage IIa (Figure 1).

Next, we examined the effect of the degree of damage to regional lymph nodes on the content of cytokines in saliva in breast cancer (Figure 2). It was shown that the maximum deviations from the control group in all cases were observed for patients with $N_2 + N_3$; all statistically significant differences were found for this subgroup (Figure 2). For TNF-α,

MCP-1, IL-18, and IL-4, the nature of the content change was the same: a decrease for the $N_0$ and $N_1$ subgroups and a sharp increase in the content for the $N_2 + N_3$ subgroup (Figure 2).

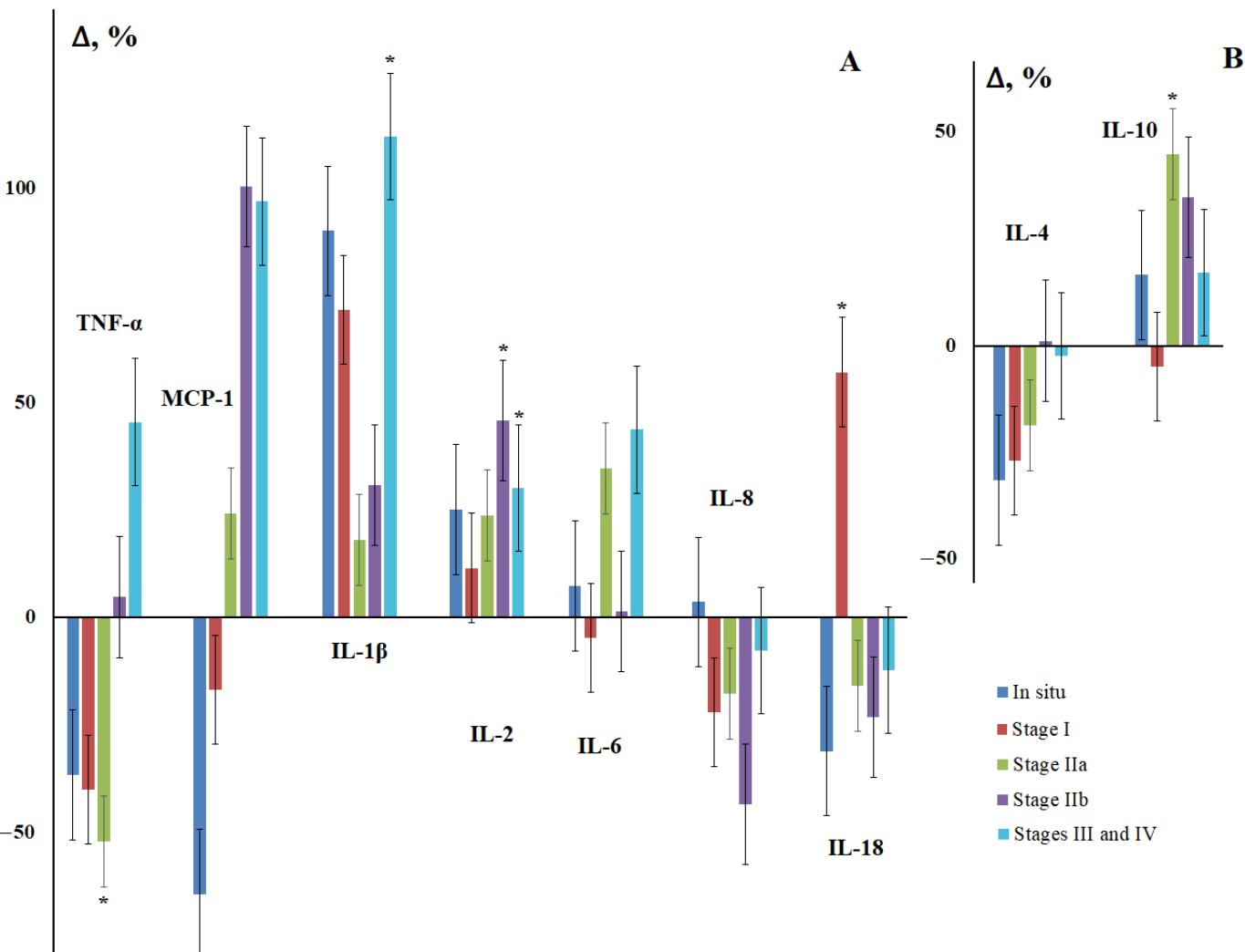

**Figure 1.** Concentration of salivary cytokines for different stages of breast cancer relative to the control group. (**A**)—pro-inflammatory cytokines, (**B**)—anti-inflammatory cytokines. Here and in further figures, the axis "0-0" represents the level of cytokines in the control group. All changes are shown relative to the control group (%). *—differences with the control group are statistically significant, $p < 0.05$.

As an important characteristic of breast cancer, the degree of tissue differentiation and its relationship with the level of salivary cytokines was considered (Figure 3). The maximum differences from the control group were observed for highly differentiated tumors, while the content of all cytokines decreased, except for MCP-1 and IL-1β (Figure 3). For poorly differentiated breast tumors, an increase in the content of IL-1β, IL-2, IL-6, and IL-10 was observed, while the content of the rest decreased compared to the control group, but to a lesser extent than for highly differentiated tumors.

At the next stage of the study, the influence of the molecular biological subtype of breast cancer on the content of cytokines in saliva was considered (Figure 4). Since there were only four patients in the non-luminal breast cancer subgroup, this subgroup was excluded from consideration and only four molecular biological breast cancer subtypes were evaluated. For the luminal A and luminal B (HER2-negative) subtypes, the nature of

the change in the content of cytokines practically coincided (Figure 4), and the maximum differences from the control group were shown for triple-negative breast cancer.

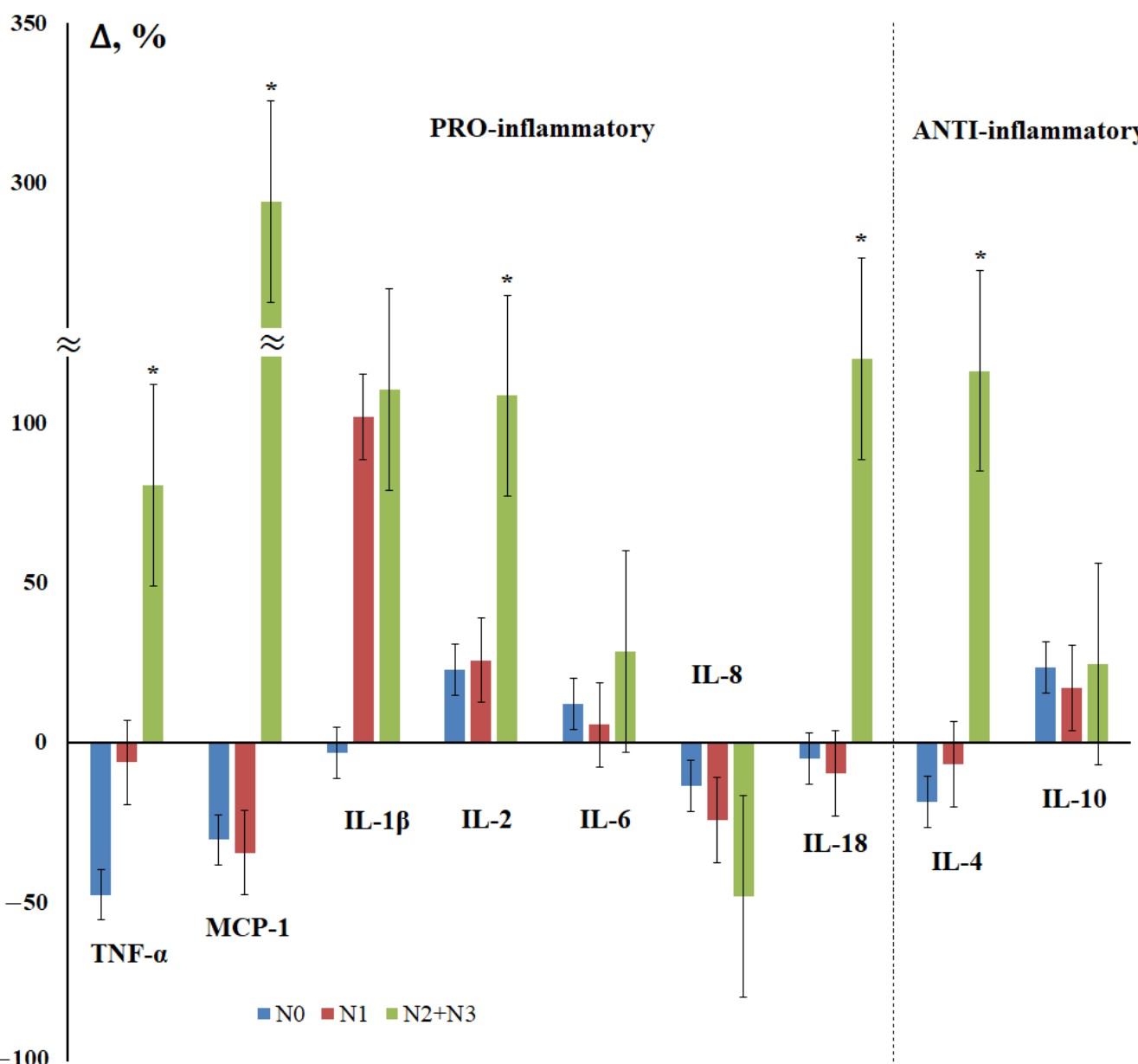

**Figure 2.** Changes in the concentration of salivary cytokines in the status of lesions of regional lymph nodes in breast cancer compared with the control group. $N_0$, $N_1$, $N_2 + N_3$—regional lymph node injury status. *—differences with the control group are statistically significant, $p < 0.05$.

According to the content of IL-1β, IL-6, IL-8, IL-18, and IL-10, the subgroup of luminal B (HER2-positive) cancer differed from the rest of the luminal subtypes, which suggested that it was HER2 status that was the determining factor influencing on the content of cytokines in saliva. Indeed, the content of pro-inflammatory cytokines differed depending on the HER2 status; the levels of TNF-α, MCP-1, IL-1β, and IL-8 changed in different directions (Figure 5a). No differences in the content of anti-inflammatory cytokines depending on the HER2 status were found.

Depending on the hormonal status of the tumor, it was noted that the maximum differences from the control group were characteristic of ER- and PR-negative tumors (Figure 5b,c), while the ER status had a greater effect on the content of cytokines in saliva.

An extreme increase in the content of MCP-1 in saliva was noted for ER-negative tumors (Figure 5b). Statistically significant differences between ER-positive and ER-negative tumors were confirmed for IL-2 and IL-4 (Figure 5b). Interesting was the fact that the level of IL-4 increased for hormone-independent tumors and decreased for hormone-dependent ones (Figure 5b,c).

The expression of Ki-67 also determined the multidirectional nature of changes in the content of IL-2, IL-18, and IL-10 in saliva (Figure 6). The maximum decrease in the content of cytokines was noted for the group with a low level of Ki-67 (TNF-α, MCP-1, IL-8, and IL-4), with the maximum increase being for the group with a high level of expression of Ki-67 (IL-1β, IL-2, IL-18, and IL-10) (Figure 6).

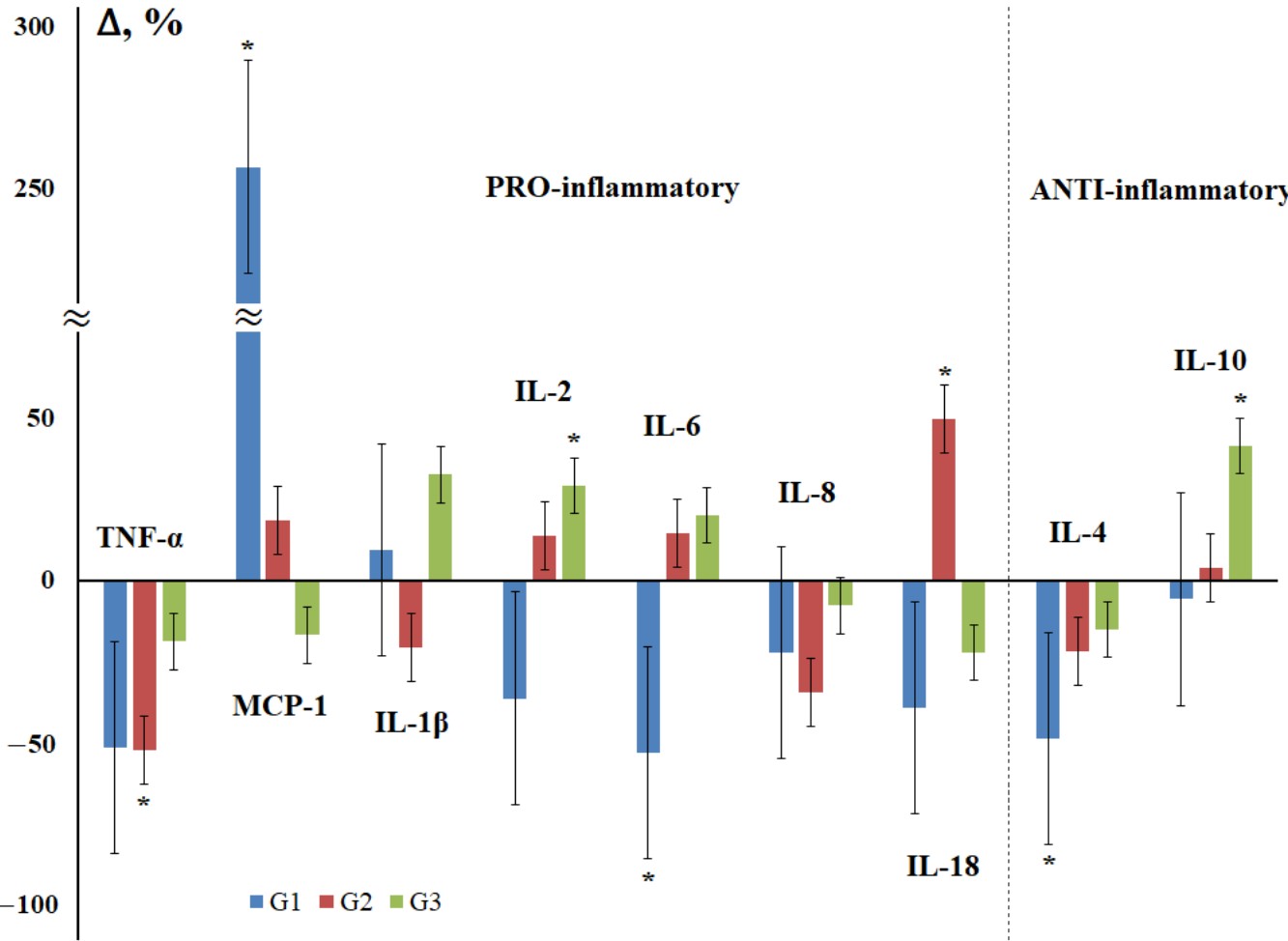

**Figure 3.** Changes in the concentration of salivary cytokines depending on the degree of tissue differentiation in breast cancer compared with the control group. G—degree of tissue differentiation: G1—highly differentiated, G2—moderately differentiated, G3—poorly differentiated cancer. *—differences with the control group are statistically significant, $p < 0.05$.

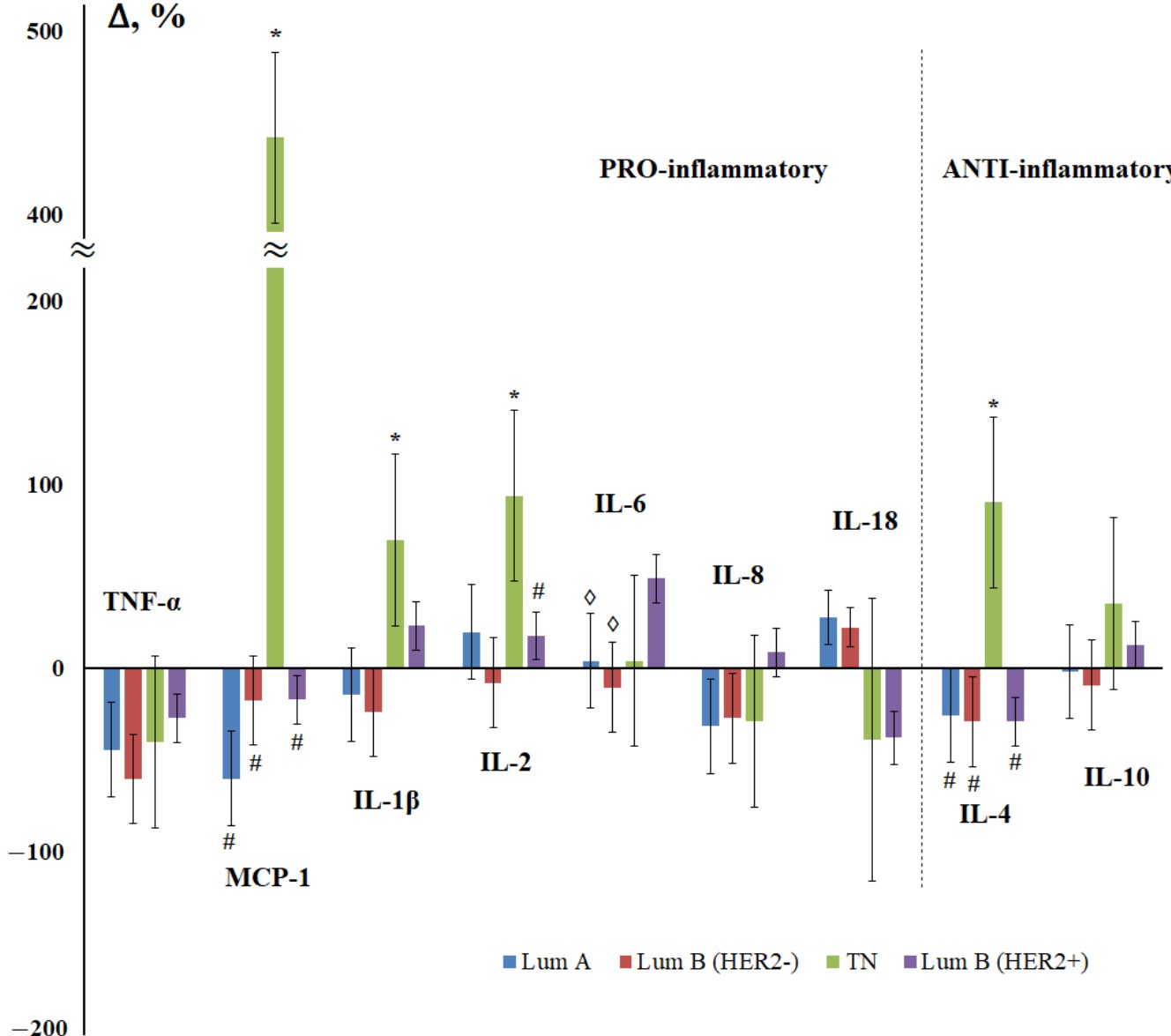

**Figure 4.** Change in the concentration of salivary cytokines depending on the molecular biological subtype of breast cancer compared with the control group. Lum A—luminal A-like, Lum B (HER2-)—luminal B-like (HER2-negative), TN—triple-negative, Lum B (HER2+)—luminal B-like (HER2-positive) breast cancer. *—differences with the control group are statistically significant; #—differences with TN are significant; ◊—differences with Lum B (HER2+) are significant, $p < 0.05$.

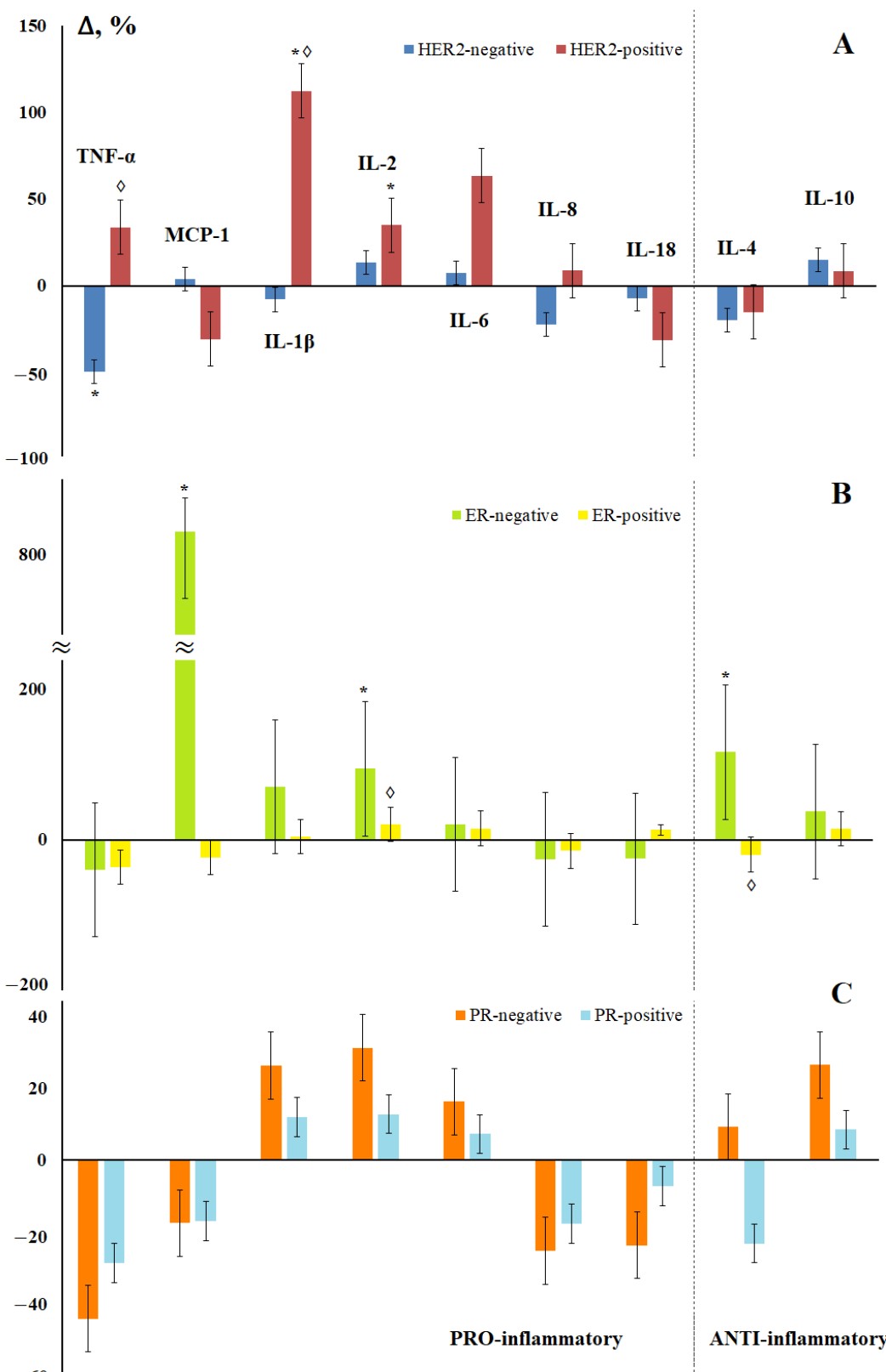

**Figure 5.** Changes in the concentration of salivary cytokines depending on the status of HER2 (**A**), estrogen (**B**), and progesterone (**C**) receptors in breast cancer compared with the control group. *—differences with the control group are statistically significant, ◊—differences between (−) and (+) are significant, $p < 0.05$.

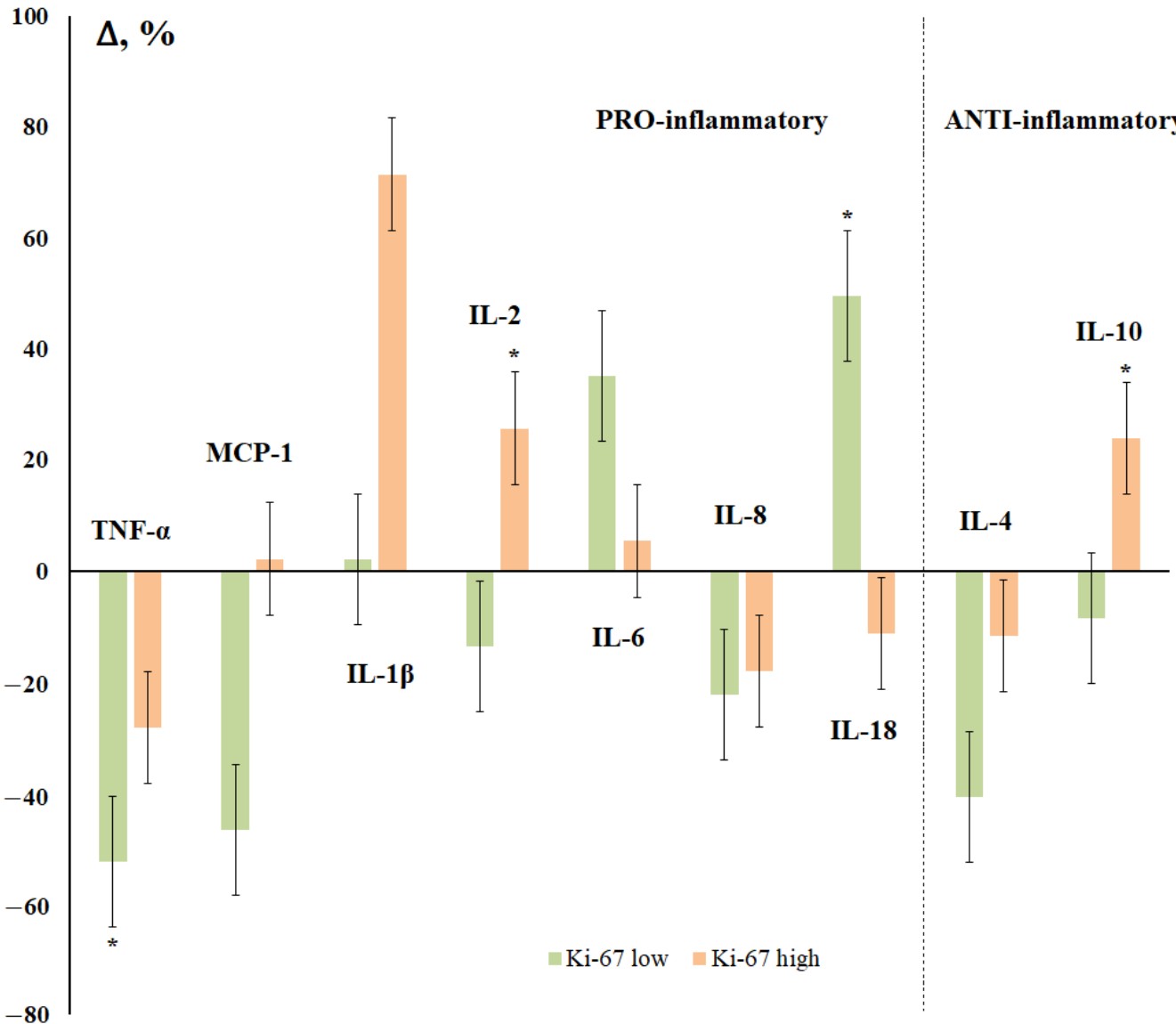

**Figure 6.** Changes in the concentration of salivary cytokines depending on the status of Ki-67 receptors in breast cancer compared with the control group. *—differences with the control group are statistically significant, $p < 0.05$.

## 4. Discussion

In recent years, it has been established that infiltrating and tumor-associated immune cells in breast cancer have both antitumor activity and tumorigenic effects [41,42]. Thus, the authors found a statistically significant increase in the production of cytokines IL-1β, MCP-1, IL-2, IL-6, IL-8, IL-10, and TNF-α (except for IL-18) in peripheral blood cells [43–45]. In saliva, the content of cytokines varies ambiguously, but this is due to a greater extent of the heterogeneity of the group and requires a detailed analysis in more homogeneous subgroups. Since the published data on the content of cytokines in the saliva of patients with breast cancer are still scarce, the obtained results were compared with the corresponding indicators for blood [3,46–52].

The cytokine IL-1β can enhance various processes that promote angiogenesis or tumor growth and the progression of breast cancer, and is considered a strong causal factor contributing to the development of malignancies, the expression of which is associated with progressive disease [53,54]. In our study, the level of IL-1β is elevated at all stages of breast cancer, particularly in the presence of even a single metastasis in the lymph nodes

(Figure 2). A significant increase was noted in HER2-positive breast cancer and with a high level of Ki-67 expression, which may be associated with a poor prognosis [55].

We have established that the level of IL-2 increases statistically significantly in the saliva of patients with breast cancer, and also increases at advanced stages of the disease (Table 1, Figure 1). A higher level of IL-2 is associated with damage to regional lymph nodes and low tumor differentiation (Figures 2 and 3), as well as HER2-positive and ER-negative tumor status and a high level of Ki-67 expression (Figures 5 and 6). This fact is consistent with the literature data, according to which a higher level of IL-2 in the blood serum of patients with breast cancer is due to its role in increasing the survival and proliferation of activated CD8+ T-cells, which in turn leads to an increase in the overall cytotoxic response [48].

By the nature of the dynamics of IL-6 almost completely repeats the patterns identified for IL-2. A distinctive feature is a higher correlation of its content with HER2-positive status and a low level of Ki-67 expression (Figures 5a and 6), which, accordingly, determines a higher level of IL-6 in saliva in luminal B (HER2-positive) breast cancer. This subtype of breast cancer is known to be more aggressive and have an overall poorer prognosis [56]. A number of studies have shown that elevated inflammatory markers in the serum of breast cancer patients, including IL-6, are associated with a poor prognosis of the disease [57,58]. IL-6, like MCP-1, is produced by mesenchymal stem cells, promoting the migration and metastasis of breast cancer cells, which can serve as one of the factors of tumor growth and progression [59].

We have found that the level of MCP-1 in the early stages of breast cancer is minimal, and then increases (Figure 1). It is known that the expression of MCP-1 (CCL2) by a primary tumor is largely associated with the percentage of macrophages and the total number of Th-cells, respectively, in the tissue of the primary tumor [60]. Interestingly, high levels of MCP-1 correlate with a higher percentage of Th2 cells compared to Th1 in primary tumor tissue and induce macrophages to secrete Th2, which recruits the chemokines CCL17 and CCL22. A higher Th2:Th1 ratio is known for a pro-tumor anti-inflammatory response. The authors showed that the expression of TGF-β is inversely proportional to the expression of MCP-1 in the early stages and is directly proportional to the expression of MCP-1 in the late stages of breast cancer, therefore, TGF-β promotes tumor progression in the late stages of breast cancer. In primary breast tumors, MCP-1 expression produces various angiogenic factors and promotes angiogenesis [61]. The expression of MCP-1 by stromal elements and the parenchymal cells of invasive human breast ductal carcinoma has been demonstrated in vivo [62]. MCP-1, derived from non-tumor stromal cells, promotes the lung metastasis of breast cancer cells [63]. According to our data, we see an increase in MCP-1 expression in case of damage to regional lymph nodes (Figure 2). The overexpression of MCP-1 causes cell invasion and metastasis, which leads to disease progression in triple-negative breast cancer [64]. In mice deficient in MCP-1, a slowdown in mammary tumorigenesis and a decrease in localized inflammation was demonstrated [65]. It was in triple-negative breast cancer that we demonstrated an increase in MCP-1 expression by more than five times (Figure 4).

For IL-8, a decrease in the content was noted in breast cancer (Table 1), and there is a decrease in its content at advanced stages, as well as with pronounced regional metastasis (Figure 1, 2). This fact is not consistent with the literature data showing that the level of IL-8 in the blood serum is higher in patients with breast cancer and may be an independent prognostic indicator for this disease [47,49]. It has been shown that the expression level of IL-8 increases in inflammatory breast cancer [66]. In our study, in the early stages of breast cancer and in fibroadenomas, the level of IL-8 reaches its maximum values; in the transition to advanced stages, the observed decrease characterizes a more pronounced immune response in early breast cancer.

Previously, we determined the content of IL-18 in the saliva of healthy women (17.4 [7.0; 41.0] pg/mL) [37]. In this work, we have shown that the level of IL-18 in saliva was equally increased in both fibroadenomas and breast cancer. Apparently, this shows that

inflammation contributes more to the growth of this parameter than tumor growth. In saliva, the level of IL-18 increases only with regional lymphogenous metastasis (Figure 2). This is in good agreement with data showing that IL-18 may be a critical factor in breast cancer metastasis [51,67–69].

An increase in the level of TNF-$\alpha$, on the one hand, accompanies the induction of apoptosis; on the other hand, it stimulates neoangiogenesis and causes an increase in the death of lymphocytes infiltrating the tumor [70]. As a result, there is an increase in the proliferation and spread of tumor cells [71]. This is in good agreement with the fact that we see an increase in the concentration of TNF-$\alpha$ in saliva in advanced stages of breast cancer and in regional lymph nodes. The co-expression of IL-6 and TNF-$\alpha$ has been associated with lymph node involvement and shorter survival [72].

For anti-inflammatory cytokines, there is a mutually opposite pattern of changes in the content in breast cancer. It is known that the level of IL-4 is higher in malignant tumors of the breast than in benign ones [50]. According to our data, the level of IL-4 in saliva remains almost constant both against the background of fibroadenomas and in breast cancer. In the early stages of the disease, there is a decrease in the content of IL-4 and active growth in the common ones, mainly due to the contribution of regional metastasis (Figure 2). Against the background of breast cancer, the level of IL-10 increases, reaching a maximum at stage IIa and gradually decreasing at advanced stages of breast cancer. The maximum increase in the level of IL-10 is only observed for poorly differentiated breast tumors.

Ki-67 is known to be a protein only expressed in proliferating cells [73]. Importantly, in breast tumors, high levels of Ki-67 have been associated with cancer progression [74], a higher risk of central nervous system metastases, and decreased overall survival and disease-free survival [75]. A high level of Ki-67 is accompanied by an increase in the concentration of IL-1$\beta$, IL-2, and IL-10, while in the group with low Ki-67, on the contrary, the level of TNF-$\alpha$, MCP-1, and IL-4 decreases with an increase in IL-6 and IL-18.

The limitations of the study are associated with a limited number of cytokines and chemokines included in the study, and an insufficient sample size. For a more correct interpretation of the data, parallel determination of cytokines and chemokines in saliva and blood plasma is required, which is planned to be done in the next stages of the study.

## 5. Conclusions

It has been shown that the content of pro-inflammatory and anti-inflammatory cytokines in saliva correlates well with the clinicopathological characteristics of breast cancer. Thus, the level of all salivary cytokines increased at advanced stages of breast cancer and at a low degree of tumor differentiation. The exception was MCP-1, which was found to be extremely high in well-differentiated breast cancer. A statistically significant increase in the content of MCP-1, IL-1$\beta$, IL-2, IL-4, and IL-10 was found in triple negative breast cancer. For the first time, the correlation of the salivary levels of TNF-$\alpha$, IL-1$\beta$, and IL-6 with HER2 status, MCP-1, IL-1$\beta$, IL-2, and IL-4 with the hormonal status of the tumor was shown. The relationship between the level of IL-2, IL-10, and IL-18 in saliva with the level of expression of Ki-67 has been established. Our results allow us to assess the prognosis of the disease and, in the future, to determine the salivary cytokine profile during treatment and to control relapse, since there is no way to regularly determine the level of cytokines in the tumor microenvironment. All of this demonstrates the promise of studying salivary cytokines in oncological diseases.

**Author Contributions:** Conceptualization, L.V.B. and A.I.L.; methodology, L.V.B.; validation, L.V.B., E.A.S. and A.I.L.; formal analysis, L.V.B.; investigation, E.A.S.; resources, A.I.L.; data curation, A.I.L.; writing—original draft preparation, E.A.S.; writing—review and editing, L.V.B.; visualization, E.A.S.; supervision, L.V.B. All authors have read and agreed to the published version of the manuscript.

**Funding:** This research received no external funding.

**Institutional Review Board Statement:** The study was conducted according to the guidelines of the Declaration of Helsinki and approved by the Ethics Committee of Omsk Regional Clinical Oncological Dispensary (21 July 2016, protocol No. 15).

**Informed Consent Statement:** Informed consent was obtained from all subjects involved in the study.

**Data Availability Statement:** The data presented in this study are available on request from the corresponding author. The data are not publicly available because they are required for the preparation of a Ph.D. thesis.

**Conflicts of Interest:** The authors declare no conflict of interest.

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
