# Peer review of "Pro-Inflammatory and Anti-Inflammatory Salivary Cytokines in Breast Cancer: Relationship with Clinicopathological Characteristics of the Tumor"

_cimb, doi:10.3390/cimb44100319_

Round 1

Reviewer 1 Report (Previous Reviewer 3)

line 343 IL-1 missing beta

There should be one sentence in the introduction about the importance of MCP-1 and IFN in breast cancer

Figure 1 illegible. It should be broken down into 9 graphs showing the level of each of the tested cytokines

The authors should show cytokine levels in patients with Triple-negative Breast Cancer. TNBC may not be basal-like. Also, basal-like doesn't always have to be TNBC.

Bertucci F, Finetti P, Cervera N, Esterni B, Hermitte F, Viens P, Birnbaum D. How basal are triple-negative breast cancers? Int J Cancer. 2008;123(1):236-40. doi: 10.1002/ijc.23518

Author Response

Reviewer 2 Report (New Reviewer)

Current issues in molecular biology (CIMB)-2022- cimb-1932641.  Pro-inflammatory and anti-inflammatory salivary cytokines in breast cancer: relationship with clinicopathological characteristics of the tumor by Bel’skaya et al.

The manuscript by Bel’skaya et al. investigated the expression of pro-inflammatory and anti-inflammatory salivary cytokines profiles in breast cancer and establish their relationship with clinicopathological characteristics of the tumor. Although the research group has similar previous works on the subject, the current one provides a control group of benign breast lesions that gives solidity to the results and their interpretations. In my opinion, it, therefore, contributes to knowledge in this area, which is still incipient and of interest. Some few minor comments are including

Minor comments:

The information given between lines 62-634 does not contribute to the current work unless explained in more detail.

Including the control group, with 11 patients with fibroadenomas, is very important, but it is unclear how the comparison was made with the group of 113 cancer patients. Were all the plotted values ​​normalized concerning the values ​​obtained for the control group? A graph between both groups would also have been useful instead of the table.

Greater detail could be included in the performance of the ELISA. For example, were the same sample volume, incubation time, reading equipment, and positive controls used as recombinant cytokine samples, serum, or control tissues for all?

In all the figures it is suggested to include as a title at the top/bottom to facilitate reading: N0, N1, N2+N3 regional lymph node injury status, etc. Another example: G1, G2, G3 degree of tissue differentiation

All legends should be reviewed and improved to provide clearer and more detailed information. In figure 3 there seems to be an error in the legend since the significantly increased cytokines are MCP-1 and IL-10 according to the graph and the statistical analyses shown.

Figure 5 includes the use of the letters A, B and C in the legend and it is suggested to change the colors of the graphs that are shown in each letter in such a way that it is clear that they are different analyzes

In the discussion of line 245, the authors contradict themselves because they indicate that there are no published data on cytokines in saliva, having a previous work and then citing another. It is suggested to modify because the works to date are "still scarce".

In lines 255 and 273 at the beginning with "It has been established" it does not make it clear that in the context of this work and not of previous evidence or of other authors, it is suggested to improve the wording.

Round 2

Reviewer 1 Report (Previous Reviewer 3)

ok

This manuscript is a resubmission of an earlier submission. The following is a list of the peer review reports and author responses from that submission.

Round 1

Reviewer 1 Report

In the manuscript entitled “Pro-inflammatory and anti-inflammatory salivary cytokines in 2 breast cancer: relationship with clinicopathological characteristics of the tumor” the authors are reporting on salivary cytokines which are altered in breast cancer subjects. Although the manuscript is somehow interesting it is not reflecting by any means on metabolism. Metabolites is a Metabolomics journal focused on metabolite alterations and metabolic dysregulations in context of physiological and pathophysiological conditions. Thus, more clinical journal for this study might be a better fit. I have following suggestions to the current manuscript.

1)      There are multiple diseases including skin disease as well as diabetes. It would be important to comment whether subjects were suffering of either skin disease or diabetes. If the HbA1C levels are available, it would be important to include in the clinical data part.

2)      Additionally, obesity is one of the factors related to breast cancer and strongly related to inflammation. The information on the BMI is missing.  

3)      It would be suggested to present the bar plots in form of the box plots containing information on the number of subjects on the plot (box overt the dot where each dot = 1 subject). It will be also informative to include the control group on the boxplot. Each cytokine can be presented on the separate graph to make it less busy in Panel A the pro-inflammatory whereas in panel B anti-inflammatory molecules can be included.

4)      The authors looked simultaneously at multiple molecules thus the 0.05 p-value should be corrected by the number of tests.

5)      It is suggested to calculate the correlation between the Ki67 and the levels of cytokines. It would be more informative than grouping the Ki67 into low and high.

Reviewer 2 Report

The paper entitled “Pro-inflammatory and anti-inflammatory salivary cytokines in breast cancer: relationship with clinicopathological characteristics of the tumor”, provides new insights into the salivary cytokine profile of patient’s with breast tumors. However, I have a few remarks.

1. In Introduction section line 43 the Authors describes a role of IL-1, it would be useful to specify its role in tumorigenesis and inflammation than to just write that it is important.

2. Line 54, that sentence needs some clarification.

3. line 57,58” IL-10 plays an important role in the pathogenesis of cancer: excessive production of IL-10 increases the likelihood of tumors due to immunosuppression” the cited research does not support that statement, however that meta-analysis does: Li L, Xiong W, Li D, Cao J. Association of Interleukin-10 Polymorphism (rs1800896, rs1800871, and rs1800872) With Breast Cancer Risk: An Updated Meta-Analysis Based on Different Ethnic Groups. Front Genet. 2022 Feb 4;13:829283. doi: 10.3389/fgene.2022.829283

4. line 59 ”IL-10 inhibits angiogenesis, and, consequently, tumor growth and metastasis [6 ,16].” Please indicate where in these two cited works it is shown that IL-10 inhibits angiogenesis? 

5. Is it possible to presented the characteristic of the studied group in different form, to make these data more transparent, perhaps some table?

6 Section 2.2 the Authors did not indicate the source of antibodies used to IHC staining in the process of the receptor status analysis of the breast cancer.

7. The authors did not present the results of correlations between the T and M feature and the salivary cytokine profiles, could the Authors explain why?

The approach to assess the molecules in saliva is very useful due to atraumatic collection of samples. The authors proved usefulness of investigating the cytokine profile and its link to  clinicopathological features of tumors, however it may be beneficial to expand the investigation for biomarkers different then cytokines. Also in conclusion section please indicate the limitations of use of the saliva cytokine profile as a biomarker in monitoring of the cancer disease. 

Reviewer 3 Report

line 36, 43 IL-1 missing beta
line 156 TNF missing alpha

There should be one sentence in the introduction about the importance of MCP-1 and IFN in breast cancer

Figure 1 illegible. It should be broken down into 9 graphs showing the level of each of the tested cytokines

The authors should show cytokine levels in patients with Triple-negative Breast Cancer

The abstract is 258 words long. It should be up to 200 words.

Reviewer 4 Report

Many thanks for the opportunity to participate.

The authors assessed the validity of comparing the salivary cytokine profile of breat cancer patients with pathology of the tumor. Authors provide a good overview and defined cytokines, as well as the mechanism of actions of cytokines. Reference was make to their role in immunology, as well as tumor proliferation and metastasis, as well as apoptosis as a form of cell death.

The research is well-designed and methodology used successfully addressed the envisaged objectives. A clear outcome for triple negative breast cancer (statistically significant) was reached pertaining to MCP-1, IL-1 β, IL-2, IL-4 and IL-10. According to the authors their research is novel and such findings have not previously been addressed in literature. Authors clearly stated that their findings largely contribute to disease prognosis.

Statistical analyses were thoroughly conducted (including the Allred Scoring Guideline).

Perhaps the authors can elaborate more in their discussion pertaining to specific research that can take this data further in the study of oncology and salivary cytokines.

I would also suggest that the authors please include more recent reference as well.

Although no external funding was needed, it would also be good to please acknowledge the type of internal funding obtained.